# Mapping Bacterial Biofilm on Features of Orthopedic Implants In Vitro

**DOI:** 10.3390/microorganisms10030586

**Published:** 2022-03-08

**Authors:** Kelly Moore, Niraj Gupta, Tripti Thapa Gupta, Khushi Patel, Jacob R. Brooks, Anne Sullivan, Alan S. Litsky, Paul Stoodley

**Affiliations:** 1Department of Microbial Infection and Immunity, The Ohio State University, Columbus, OH 43210, USA; kelly.e.moore20@gmail.com (K.M.); niraj.gupta@osumc.edu (N.G.); tripti.gupta@osumc.edu (T.T.G.); jacob.brooks@osumc.edu (J.R.B.); 2College of Public Health, The Ohio State University, Columbus, OH 43210, USA; patel.3731@osu.edu; 3Department of Orthopedics, The Ohio State University, Columbus, OH 43203, USA; anne.sullivan@osumc.edu (A.S.); litsky.1@osu.edu (A.S.L.); 4Department of Biomedical Engineering, Ohio State University, Columbus, OH 43210, USA; 5Department of Microbiology, The Ohio State University, Columbus, OH 43210, USA; 6National Centre for Advanced Tribology at Southampton (nCATS), National Biofilm Innovation Centre (NBIC), Department of Mechanical Engineering, University of Southampton, Southampton SO17 1BJ, UK

**Keywords:** biofilm attachment, biofilm mapping, implant-associated infection, orthopedic biomaterials, periprosthetic joint infection, surface roughness

## Abstract

Implant-associated infection is a major complication of orthopedic surgery. One of the most common organisms identified in periprosthetic joint infections is *Staphylococcus aureus,* a biofilm-forming pathogen. Orthopedic implants are composed of a variety of materials, such as titanium, polyethylene and stainless steel, which are at risk for colonization by bacterial biofilms. Little is known about how larger surface features of orthopedic hardware (such as ridges, holes, edges, etc.) influence biofilm formation and attachment. To study how biofilms might form on actual components, we submerged multiple orthopedic implants of various shapes, sizes, roughness and material type in brain heart infusion broth inoculated with *Staphylococcus aureus* SAP231, a bioluminescent USA300 strain. Implants were incubated for 72 h with daily media exchanges. After incubation, implants were imaged using an in vitro imaging system (IVIS) and the metabolic signal produced by biofilms was quantified by image analysis. Scanning electron microscopy was then used to image different areas of the implants to complement the IVIS imaging. Rough surfaces had the greatest luminescence compared to edges or smooth surfaces on a single implant and across all implants when the images were merged. The luminescence of edges was also significantly greater than smooth surfaces. These data suggest implant roughness, as well as large-scale surface features, may be at greater risk of biofilm colonization.

## 1. Introduction

Implant-associated infections (IAI) are a major complication in orthopedic surgery. While these infections occur in 0.5–2% of patients 2 years following surgery [1,2,3], the procedures are complex and are associated with substantial morbidity and economic cost [4]. With hip and knee arthroplasty procedures expected to rise 667% in the US from 2003 to 2030, the number of infections from these procedures is also expected to rise [5].

One of the most common organisms identified in periprosthetic joint infections is *Staphylococcus aureus* [6,7,8]. *S. aureus* is a Gram-positive bacterium, notorious for causing chronic infections because of many factors, including the ability to form biofilms [9,10,11,12]. Bacterial biofilms are difficult to treat as they often require higher doses of antibiotics, concentrations 1000–1500 times higher than what is needed to kill planktonic infections [13]. Bacterial biofilms are associated with 65% of implant-related infections [14].

Orthopedic implants are composed of various materials that include surfaces likely to be colonized by bacterial biofilms. It has been noted that biofilm likes to attach to titanium alloy discs, polymethyl methacrylate (PMMA), ultra-high molecular weight polyethylene (UHMW-PE), stainless steel (SST), and aluminum [15,16]. In vitro results have shown hydroxyapatite (HA) and polyethylene (PE), materials with rougher surfaces, had a greater surface area coverage than titanium and 316L SST [17]. To prevent the buildup of bacterial biofilm on these surfaces, various methods have been studied, including coating implants and electrical conduction, however, coating with antibiotics risks selecting for drug-resistant species, coating titanium–aluminum–niobium metal alloy with silver only has limited effects, and no in vivo data has been produced on electrical conduction, as no one is willing to participate in the experiment [18,19,20,21].

Mouse models have found that an implanted material required a smaller inoculum to produce a similar sized infection at the same site without an implanted material [22]. Knowing this means it is even more important to find ways to reduce biofilm attachment to hardware by first mapping where the biofilm attaches. Many studies have looked at the attachment of *S. aureus* biofilms to various materials and have found that certain materials may promote infection [23]. However, little is known about how larger surface features of orthopedic hardware (such as ridges, holes, edges, etc.) influence biofilm formation, or if certain parts of a prosthesis may be more vulnerable to biofilm accumulation. Therefore, an important step in this process is to understand the areas of real implants where bacteria may preferentially attach, and where biofilms may accumulate. In this study, we have addressed this by mapping the biofilm growth on multiple orthopedic implants with different macroscopic surface features and roughness.

## 2. Materials and Methods

### 2.1. Bacteria

A bioluminescent derivative strain of *Staphylococcus aureus* USA300 MRSA (SAP231) was used in this study [24]. A single SAP231 colony from a streaked brain heart infusion (Becton, Dickinson and Company, Sparks, MD, USA) agar plate was used to inoculate 20 mL of BHI broth for each experiment. The culture was grown overnight at 37 °C under shaking conditions (200 RPM).

### 2.2. Implants

Implants with representative surface features from a range of unused implants of various materials, shapes and sizes were collected from a wide variety of sources as donations from various vendors, including manufacturer’s samples, extras from prior projects and sales rep donations for teaching demonstrations for this study. These implants include:Total Knee System (knee)Sliding Hip Screw (ss)5.5 in by 0.5 in Fixation Plate (fix (5))3.25 in by 0.4 in Fixation Plate (fix (3))Femoral Hip Stem (fhs)Hip Replacement Ball Joint (bjh)Modular Stem (mod)7.3 MM TI cannulated screw 32 MM thread/45 MM (screw)

The abbreviations shown in parentheses are used throughout the main text and figures.

### 2.3. Growing Biofilm on Implants

Implants were placed in an autoclavable sterile Pyrex^®^ baking dish (8 in × 5 in × 4 in) followed by 400–600 mL of BHI broth or until all implant surfaces were covered. An overnight culture of SAP231 was added to the dish at a ratio of 1:1000 and mixed throughout for 30 s. The media and implants were covered and incubated for 24 h at 37 °C under static conditions. Media was exchanged every 24 h for 3 days. On the third day, the media was exchanged and incubated for 30 min before washing each implant with 10 mL of phosphate-buffered saline (PBS) to dislodge the non-adherent bacteria.

### 2.4. IVIS Imaging

An in vitro imaging system (IVIS 100, Xenogen, Waltham, MA, USA) was used to measure the metabolically active biofilm semi-quantitatively on each implant. This system uses a color scale heat map to indicate the different intensities of metabolic activity, where red is equivalent to high metabolic activity and blue or black is equivalent to low metabolic activities. An image editing software (Adobe Photoshop ver. 22.4.2) was used to stitch and overlay the colored image with a plain image to be able to visually see where the bacteria attached.

### 2.5. Scanning Electron Microscopy (SEM)

Before imaging each implant by scanning electron microscopy (SEM) (Apreo FEG SEM, Thermo Scientific, Waltham, MA, USA), each implant was soaked in prefixing agents containing 2.5% glutaraldehyde in 0.2M of cacodylate buffer (pH 7.4) for 24 h at room temperature. These components were then rinsed three times with cacodylate buffer. After the final rinse, the components were dehydrated by placing in increasing concentrations of an ethanol series (70%, 90%, and 100%) three times each for 20 min, followed by dehydration in 100% hexamethyldisilazane (HMDS) twice for 5 min each. Components were then dried at room temperature before imaging. The acquired images were then false-colored using photo editing software (Adobe Photoshop^®^ ver. 22.4.2) to distinguish the bacterial attachment from the implant surfaces. Because SEM is time-consuming over larger areas, such as implants, we chose specific regions of what we believed were areas of interest during imaging. Moreso, accurate measurement of the roughness on raised or sharp edges was not possible using the profilometer. Therefore, we restricted our measurements only to areas where the measurement was possible.

### 2.6. Roughness Characterization

The roughness of these implants at various locations exhibiting different bioluminescence intensities in IVIS images was measured using a Zeta 20 optical profilometer (KLA Corp., Milpitas, CA, USA), and the Ra value was calculated.

### 2.7. Cleaning Explants

After each use, implants were cleaned using a previously described method [25]. A 1:1000 dilution of micro-90 cleaning solution (Andwin Scientific, Schaumberg, IL, USA) and reagent grade water was created. Used implants were placed into the solution and sonicated for 30 s. The solution was removed, and more reagent grade water was added until the implants were covered. They were sonicated again for 30 s. This process was repeated until no more bubbles appeared to confirm the micro-90 solution was properly washed away. Implants were then removed, dried and autoclaved. These cleaned implants were imaged with SEM to confirm the efficacy of the cleaning procedure, and the images are included as Appendix A. Furthermore, we confirmed the absence of any organic or soap-like materials after the cleaning process, using Raman spectroscopy (Appendix A). Raman spectroscopy has been previously used to identify soap residues on surfaces that show distinct peaks [26].

### 2.8. Quantification

The freely available NIH Image J [27] was used to measure the mean luminescence from captured grayscale IVIS images. The scale was initially set to 308.0833 pixels/85.5 mm. Four different sampling locations representing the various features of each implant were pre-selected to measure the luminescence. In addition, 30-pixels circles in diameter (55.15 mm^2^) were measured and recorded.

### 2.9. Statistics

Two-way ANOVA was conducted for each explant using GraphPad Prism, version 8.4 for Windows (GraphPad Software, San Diego, CA, USA). The locations on each implant were also divided into groups based on the types of surfaces (rough, edges, smooth) and a one-way ANOVA was conducted comparing these groups. Differences were considered statistically significant for *p* < 0.05.

## 3. Results

The representative implants’ surface roughness without any bacterial growth, as measured by the optical profilometry, are shown in Figure 1.

*Staphylococcus aureus* SAP231 attachment to various implants after three-day incubation was observed via SEM and IVIS imaging. The SEM micrographs are shown in Figure 2. It is evident that the surfaces with higher roughness values had more bacterial attachment when compared to the smooth surfaces. Different magnifications were used to capture the different scales of bacterial colonization with SEM. The relationship between the surface and the aggregates could not be visualized if a higher magnification was used, as the background surface could not be seen. In contrast, the coccal form could not be seen if we used the lower magnification for the sparse covering of cells.

The IVIS images in Figure 3 show the bioluminescent signal superimposed on the photographic image to show the implants with circles representing different areas of roughness and features on the implants. A and E in Figure 3 look colored because of the presence of a more bioluminescent signal; however, even though many of the circles look dark by eye, there were detectable signals from the grayscale image. The average luminescence of each circle was then quantified and presented as the geometric mean of the Log10 transformed luminescence with SE bars (Figure 4). Due to the order of magnitude of differences between implants, the graphs in Figure 4 were plotted to highlight the variations from location to location on an individual implant with a logarithmic scale.

Across all implants, the rough surfaces (Ra ≥ 17 µm) indicated in red had the greatest luminescence compared to edges (green) and smooth surfaces (blue) (Figure 5). The luminescence quantified from the rough surfaces of the total knee and the femoral hip stem was significantly greater than the edges and/or smooth surfaces of those implants (Figure 4A,E) (*p* < 0.05). Edges had a greater luminescence compared to smooth surfaces across all implants. The sliding screw is one example of this, as the luminescence from two of the three edge locations was significantly greater than the smooth surface (*p* < 0.05).

When merging the average luminescence of each surface type together, the luminescence of the rough surfaces was significantly greater than the luminescence of edges and smooth surfaces (Figure 6) (*p* < 0.0001). The luminescence of the edges was significantly greater than the smooth edges (*p* < 0.0001).

## 4. Discussion

*S. aureus* is one of the most common pathogens of periprosthetic joint infections. Although various options have been tried to prevent these infections from occurring, i.e., coating, using antibiotic therapy, etc., these treatments have not eliminated them. In this study, we set out to map where implant-associated infections may begin to further understand the attachment to hardware.

We observed greater attachment to rough surfaces on implants, such as the femoral hip stem and the total knee system, than smooth-like surfaces found elsewhere (Figure 4 and Figure 5). Although this is not surprising, as greater surface area attachment to rougher surfaces and lesser surface area attachment to smooth surfaces is reported across many fields [17,28,29,30]. Many explain this through the attachment point theory, which describes how increased contact points result in a greater and stronger attachment, and additionally, through the shelter effect in which the bacteria are protected, such as a form of shelter in the valleys of the rough surface [31,32,33]. SEM imaging, a commonly used technique to identify biofilm on orthopedic materials [34], demonstrates this more clearly through the increased points of attachment and valleys on the implants we characterized as rough compared to the smooth or edge pieces (Figure 1 and Figure 2). A similar study from Luca et al. reported a reduction in bacterial attachment to polished surfaces compared to other surfaces on spinal implants [23]. We found similar results in our study with a significant reduction in bacterial attachment to smooth implant surfaces (Figure 6).

There was a significant bacterial attachment to edges on implants (Figure 6). One example is the 5-inch fixation plate location 1, where there is a depression used to identify the component (Figure 3). This location had the greatest luminescence of the four locations on the 5-inch fixation plate, three of which are characterized as edges. Knee location 4 and sliding screw location 2 are similar examples. The reasoning behind this attachment and biofilm accumulation on edges is unclear. In industrial systems with flow, it is known that biofilms attach and accumulate on macroscopic edges and ridges that might be found in bends and joints [35] and it is assumed that this is due to vortices and back eddies around these features. However, in our laboratory experiments, there was only flow during media exchanges and removal of implants from the growth media. An edge may allow delivery of nutrients from both sides of the biofilm, encouraging growth. More experiments are required to test this hypothesis. Nevertheless, our study shows that macroscopic surface features, such as edges, ridges, tapped screw holes, etc., as well as macroscopically rough surfaces, such as those designed for osseointegration, should be considered as potential sites for biofilm accumulation. Conventionally, bacterial attachment studies are performed on flat featureless coupons with uniform nano or micro surface roughness and such macroscopic features are overlooked.

We acknowledge several limitations of our study. First, we only used one species, *S. aureus*. This allowed us to only observe the attachment to various implant surfaces of a species-specific adhesion method, specific size and specific morphology. The cocci shape of *S. aureus* may also allow attachment to a different set of crevices than a larger or rod-shaped bacterium. In addition, while we took several measures to clean each implant between experiments, they were not as clean as a new unused implant. Debris that was left behind may have encouraged repeated attachment to an area that otherwise would not have had a significant attachment. In addition, the level of bioluminescence is a function of not only numbers of bacteria but also metabolic activity and cannot differentiate by light alone. Therefore, we used SEM to corroborate that the areas which were emitting more light, indeed, had denser bacteria. Furthermore, there is a lack of evidence to suggest that the attached bacteria in this study are in the biofilm phenotype, although SEM images do show bacterial aggregates and other smaller aggregates similar in appearance, as seen clinically [36,37]. Lastly, it is possible that the planktonic cells were not completely washed from the implants, leaving pools of cells in the screw holes and thus increasing the luminescence found on edge surfaces.

## 5. Conclusions

We found that surface roughness is an important factor that favors bacterial attachment and biofilm growth. However, larger surface features that are normally not considered for in vitro testing are also important. The findings in this article could provide insights into biofilm attachments on various topographies of an implant that would assist in designing implantable hardware that can prevent bacterial attachments. For example, any sharp edges, ridges, and unnecessary screw holes should be avoided from a biofilm perspective. Further in vivo studies need to be completed to confirm these in vitro results.

## Figures and Tables

**Figure 1 microorganisms-10-00586-f001:**
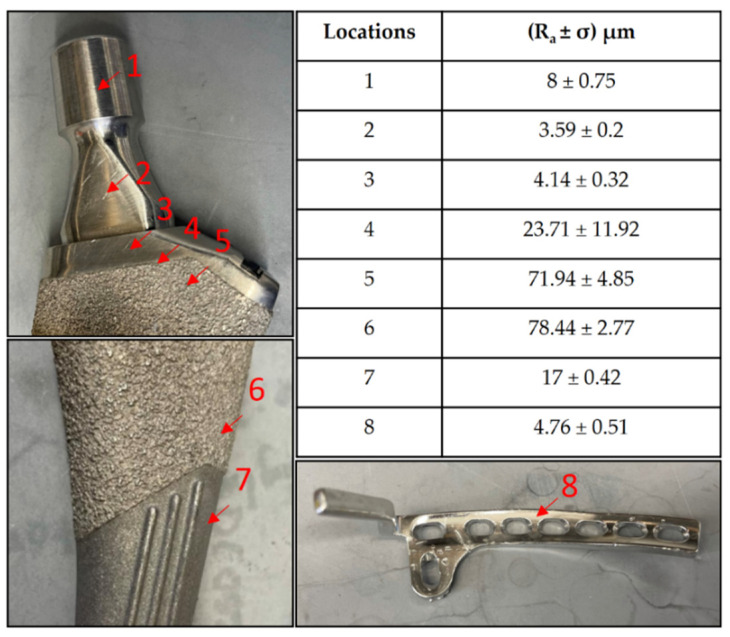
Roughness values at different surfaces on a femoral hip stem and fixation plate.

**Figure 2 microorganisms-10-00586-f002:**
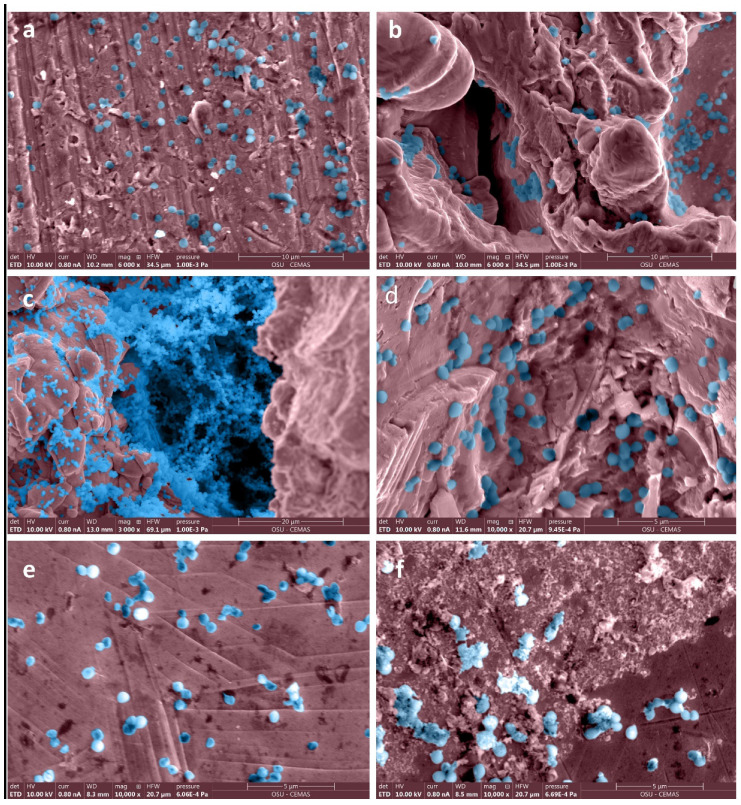
False-colored SEM images of bacteria adhered to surfaces with different roughness on femoral hip stem and fixation plate. Roughness values (Ra ± σ in µm) on surfaces of a femoral stem were (**a**) 4.14 ± 0.32, (**b**) 71.94 ± 4.85, (**c**) 78.44 ± 2.77, (**d**) 17 ± 0.42, and of fixation plate (**e**,**f**) were 4.76 ± 0.51. Original grayscale SEM images are included as Appendix A.

**Figure 3 microorganisms-10-00586-f003:**
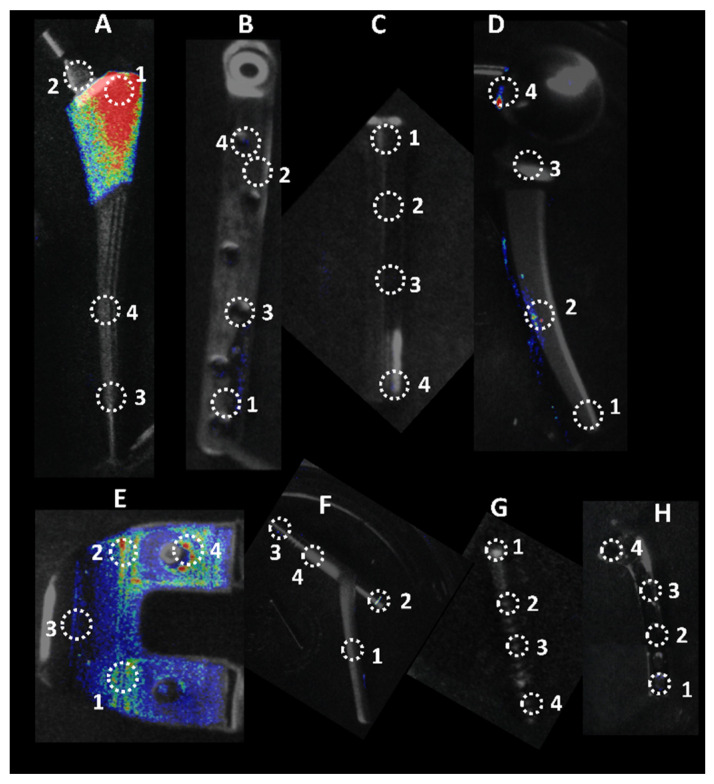
Locations of luminescence measurements. Circle diameters of 30 × 30 pixels were used to measure luminescence on eight different implants. Each location represents a different feature on the implant. A. Femoral Hip Stem; B. 5in Fixation Plate; C. Modular Stem; D. Hip Replacement Ball Joint; E. Total Knee System; F. Sliding Hip Screw; G. Ti Screw; H. 3in Fixation Plate.

**Figure 4 microorganisms-10-00586-f004:**
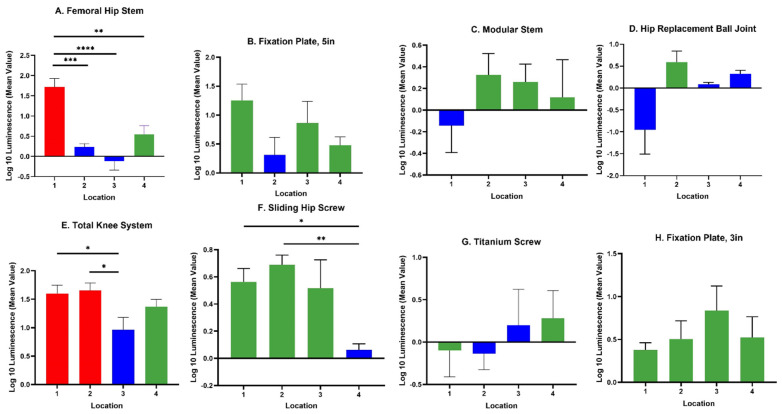
Luminescence quantification. Log 10 mean luminescence of four different locations on each implant. Red indicates rough surfaces, green indicates edges, and blue indicates a smooth surface. Geometric Average ± SE, *n* ≥ 3. (* *p* < 0.05, ** *p* < 0.01, *** *p* < 0.001, **** *p* < 0.0001).

**Figure 5 microorganisms-10-00586-f005:**
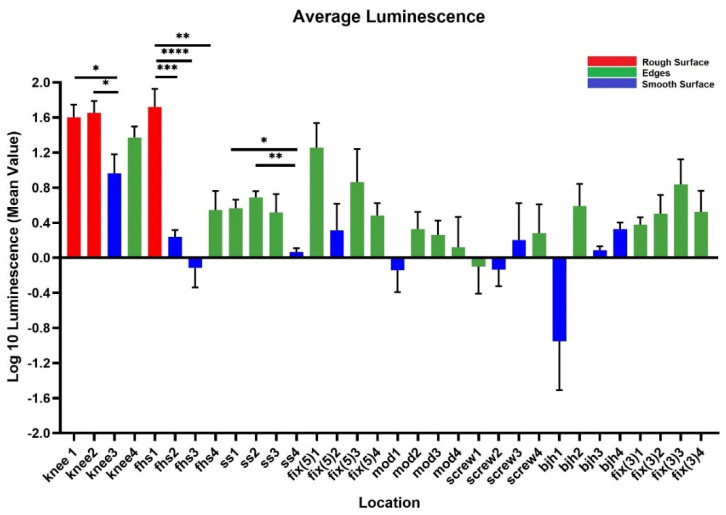
Compilation of mean luminescence of all implants. Individual graphs of each implant were combined to observe the trends across all implants. *n* ≥ 3; geometric mean ± SE. (* *p* < 0.05, ** *p* < 0.01, *** *p* < 0.001, **** *p* < 0.0001).

**Figure 6 microorganisms-10-00586-f006:**
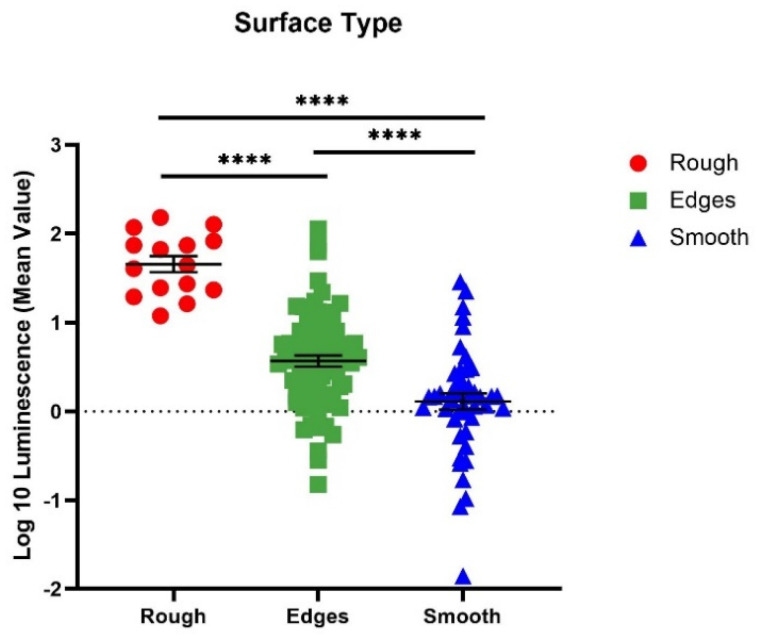
Comparison of each different surface type amongst all implants. Implant locations of similar surface types were combined to observe the average luminescence of each surface type. *n* ≥ 3; geometric mean ± SE (**** *p* < 0.0001).

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
