# Peer review of "Mapping Bacterial Biofilm on Features of Orthopedic Implants In Vitro"

_microorganisms, 2022, doi:10.3390/microorganisms10030586_

Round 1
Reviewer 1 Report
Dear authors,
I have read the short communication "
Mapping Bacterial Biofilm on Features of Orthopedic Implants 2
In Vitro".
The topic is very appropriate with current times and it is nicely written. Hence I am recommending to accept this manuscript after minor revisions.
- Abstract: I would like to suggest authors to delete the sentence "Roughness was a major factor in the colonization of bacterial biofilms" because unnecessary
- I would like to suggest authors introduce something about nanomaterial as emerging material for Orthopedic implants.
Author Response
Responses to reviewer 1 comments
Reviewer’s comment: Abstract: I would like to suggest authors to delete the sentence "Roughness was a major factor in the colonization of bacterial biofilms" because unnecessary
Authors’ Response: We agree, and we have now removed this sentence from the abstract.
Reviewer’s comment: I would like to suggest authors introduce something about nanomaterial as emerging material for Orthopedic implants.
Authors’ Response: We would like to thank the reviewer for the suggestion, and we wanted to include nanomaterials in our text, however, we could not find an appropriate place to include the information. However, we will certainly address this emerging field in future work.
Reviewer 2 Report
The authors address how bacterial attachment is occurring on various locations of titanium implants, a very important aspect in implant-related infections, using bioluminescent bacteria and SEM. Rough surfaces and edges are prone for bacterial attachment.
General comments
- Biofilm aspects like EPS and ultrastructure are not taking into account in the manuscript. How sure are the authors we look at biofilm and not single bacteria only? The SEM indicates small agglomerates of single bacteria.
- A few controls seem missing: did the authors look at bare flat material substrate of an implant under controlled conditions to show a staphylococcus biofilm? Same for controls of the implants without bacteria. And why are SEM pictures of the implants after cleaning not included (g. in supplementary data)? The latter is especially important due to the discussion of contaminations mentioned by the authors. This would the minimum data needed to confirm that what is seem in the SEM images can be claimed as bacterial biofilms.
- The authors mention that the aim is to “designing implantable hardware that can prevent bacterial attachments’’. This a very general statement. Could the authors please elaborate on how to specifically address the various and critical regions of an implant and how especially biofilm growth can be prevented?
Materials & Methods
- 2: How have the implants been selected?
- 3: Why is BHI used as medium? Might that have an influence on the bacterial attachment? Did the authors consider the addition of human serum or plasma to mimic the natural environment?
- 3: Is only 10 mL PBS used for washing the implants? Is that to dislodge the non-adherent bacteria before imaging? Please indicate.
- 4: The authors equal biomass to living (metabolically active) bacteria, but the ratio biomass to living bacteria can be different due to the amount of EPS present/produced. Did the authors confirm these statement by quantitative culture? This important information is missing from the manuscript.
- 6: Please explain why the implants are cleaned? Are the implants reused for imaging? Did the authors confirm that the implants were clean before reusing them? Bacterial debris might be the a good substate for biofilm formation.
- 7: Soap is mentioned in the last sentence. Is that part of the cleaning solution? And how is checked that the soap was sufficiently removed?
- 8: Specify how the sampling locations were selected.
- 9: What post-test has been used (Turkey, Bonferroni etc.)? Which program has been used for the statistical analyses?
Results
- Figure 1: Why are these specific regions of the implants chosen? And why not all areas or for instance the edge between 1 and 2 or between 6 and 7 or the raised brim near 7?
- Figure 2: The figures can not be compared easily, please use images with same magnification. If higher magnification is needed, use in all case. It’s an artificially colored SEM image, and that’s a very nice visual effect. However, it is not convincing how the authors chose what is a bacterium and what not. How do the authors control for this (g. quantitative culture)?
- Figure 3: The luminescent staphylococci are used commonly, however these are 2-dimentional graphs saying something about metabolism. So, you cannot tell whether a higher signal it due to a higher metabolism of one single bacterium or just more bacteria present. There is no information about the 3-dimentional structure. How is correct for that issue?
- Figure 3: Why do only A and E have that color? Explain how the places of the spots have been selected. Are the circles selected for the highest signal? (if so, why?)
- Figure 4: Use the same scale for all figures, it’s easier to compare the different implant locations in that way. Could you explain why a negative mean value can be observed? Is the luminescence compared to a control (what one?)?
- Figure 6: Is chemistry of the implants taken into account? Now all information is put on one pile, and the information on the type of material/chemistry is missing.
Discussion
- The authors focus on initial bacterial attachment. What does surface roughness mean for biofilm structure and possible entering/attachment points for EPS? Did the authors look at different time points (to be able to differentiate between initial attachment and biofilm formation)?
- The authors state “more surprising is the significant attachment to edges of implants”, however that is seen actually with every biofilm experiment, even within simple in vitro experiments where the edges of the wells are the most covered by biofilm.
Conclusion
- The authors mention the influence of various topographies, but actually they did not research the exact topography of the implants, but only looked at roughness. Could the authors be more specific in what should be done to improve the design of novel implants?
Round 2
Reviewer 2 Report
No comments